# Molecular Characterization of WCK 5222 (Cefepime/Zidebactam)-Resistant Mutants Developed from a Carbapenem-Resistant *Pseudomonas aeruginosa* Clinical Isolate

Xiaolei Pan,[a] Xinrui Zhao,[a] Yuqin Song,[b] Huan Ren,[a] Zhenyang Tian,[c] Qi'an Liang,[a] Yongxin Jin,[a] Fang Bai,[a] Zhihui Cheng,[a] Jie Feng,[b] Weihui Wu[a]

[a]State Key Laboratory of Medicinal Chemical Biology, Key Laboratory of Molecular Microbiology and Technology of the Ministry of Education, Department of Microbiology, College of Life Sciences, Nankai University, Tianjin, China
[b]State Key Laboratory of Microbial Resources, Institute of Microbiology, Chinese Academy of Sciences, Beijing, China
[c]Key Laboratory of Systems Microbial Biotechnology, Tianjin Institute of Industrial Biotechnology, Chinese Academy of Sciences, Tianjin, China

**ABSTRACT** WCK 5222 (cefepime/zidebactam) is a $\beta$-lactam/$\beta$-lactamase inhibitor combination that is effective against a broad range of highly drug-resistant bacterial pathogens, including those producing metallo-$\beta$-lactamase. In this study, we isolated a multidrug-resistant *Pseudomonas aeruginosa* clinical strain that is resistant to a variety of $\beta$-lactam antibiotics and the ceftazidime-avibactam combination. A metallo-$\beta$-lactamase gene $bla_{DIM-2}$ was identified on a self-transmissible megaplasmid in the strain, which confers the resistance to $\beta$-lactam antibiotics, leaving WCK 5222 potentially one of the last treatment resorts. *In vitro* passaging assay combined with whole-genome sequencing revealed mutations in the *pbpA* gene (encoding the zidebactam target protein PBP2) in the evolved resistant mutants. Among the mutations, a V516M mutation increased the bacterial virulence in a murine acute pneumonia model. Reconstitution of the mutations in the reference strain PAO1 verified their roles in the resistance to zidebactam and revealed their influences on cell morphology in the absence and presence of zidebactam. Microscale thermophoresis (MST) assays demonstrated that the mutations reduced the affinity between PBP2 and zidebactam to various extents. Overall, our results revealed that mutations in the *pbpA* gene might be a major cause of evolved resistance to WCK 5222 in clinical settings.

**IMPORTANCE** Antibiotic resistance imposes a severe threat on human health. WCK 5222 is a $\beta$-lactam/$\beta$-lactamase inhibitor combination that is composed of cefepime and zidebactam. It is one of the few antibiotics in clinical trials that are effective against multidrug-resistant *Pseudomonas aeruginosa*, including those producing metallo-$\beta$-lactamases. Understanding the mechanisms and development of bacterial resistance to WCK 5222 may provide clues for the development of strategies to suppress resistant evolvement. In this study, we performed an *in vitro* passaging assay by using a multidrug-resistant *P. aeruginosa* clinical isolate. Our results revealed that mutations in the zidebactam target protein PBP2 play a major role in the bacterial resistance to WCK 5222. We further demonstrated that the mutations reduced the affinities between PBP2 and zidebactam and resulted in functional resistance of PBP2 to zidebactam.

**KEYWORDS** *Pseudomonas aeruginosa*, WCK 5222, zidebactam, PBP2

Address correspondence to Weihui Wu, wuweihui@nankai.edu.cn.

The authors declare no conflict of interest.

$P$seudomonas aeruginosa is an opportunistic Gram-negative pathogen that causes various acute and chronic infections in humans (1, 2). The carbapenem-resistant mechanisms of *P. aeruginosa* mainly include reduced outer membrane permeability (largely due to mutations in the *oprD* gene), enhanced multidrug efflux system, altered

**TABLE 1** Bacterial susceptibilities to antibiotics[a]

| Strain | MIC (μg/mL) | | | | | | | | | |
|---|---|---|---|---|---|---|---|---|---|---|
| | WCK 5222[b] | ZID | FEP | CAZ | CZA | MEM | TB | GM | CIP | PB |
| NKPa-71 | 8 | 8 | 256 | 128 | 128 | 16 | 64 | 128 | 16 | 0.25 |
| PAO1 | 4 | 4 | 1 | 2 | 0.5 | 0.5 | 1 | 2 | 0.25 | 0.5 |
| PAO1/pUCP24-$bla_{DIM-2}$ | 4 | 4 | 16 | 64 | 16 | 64 | 1 | ND | 0.5 | 0.5 |
| PAO1/pXP40m | 4 | 4 | 32 | 32 | 32 | 8 | 8 | ND | 4 | 0.5 |

[a]MIC, minimum inhibitory concentration; ZID, zidebactam; FEP, cefepime; CAZ, ceftazidime; CZA, ceftazidime-avibactam; MEM, meropenem; TB, tobramycin; GM, gentamicin; CIP, ciprofloxacin; PB, polymyxin B; ND, not determined.
[b]The provisional breakpoints of WCK 5222 are 8 μg/mL or less (susceptible) and 16 μg/mL or more (resistant) (15).

transpeptidases, and acquisition of β-lactamases that are able to degrade carbapenem antibiotics (carbapenemases) (3).

β-Lactam-β-lactamase inhibitor (BLBLI) combinations have been successfully used to retain the antimicrobial activities of β-lactam antibiotics and extend the spectrum of activities against β-lactamase-producing bacteria (3, 4). However, no β-lactamase inhibitor in clinical use can inhibit metallo-β-lactamases (MBLs) (3, 4). WCK 5222 is a BLBLI combination under clinical development. It is composed of a fourth generation cephalosporin antibiotic cefepime and a β-lactamase inhibitor zidebactam. Cefepime targets PBP1a, PBP1b, and PBP3 (5). Zidebactam is a diazabicyclooctane (DBO) with dual inhibitory functions (6). It inhibits class A and class C β-lactamases. Meanwhile, zidebactam directly targets PBP2, through which it displays antibacterial activity against both *P. aeruginosa* and Enterobacterales (7–9). Thus, zidebactam functions as an enhancer for β-lactam antibiotics, whereby WCK 5222 is effective against MBL producing *P. aeruginosa* (10–13).

Recently, Barceló et al. reported a study on the *in vitro* evolution of cefepime, zidebactam, and cefepime/zidebactam resistance in the reference strain PAO1 (14). Since WCK 5222 is most likely to be used to treat infections caused by carbapenem-resistant bacteria, we examined the evolution of a multidrug-resistant clinical isolate that is resistant to meropenem and the BLBLI combination ceftazidime-avibactam, owing to the presence of an MBL gene *blaDIM-2* in a megaplasmid. We demonstrated that the evolved resistance to WCK 5222 was mainly due to mutations in the PBP2 coding gene *pbpA*. We further examined the consequences of the mutations on bacterial growth and virulence and elucidated the molecular mechanisms that confer the resistance.

## RESULTS

**Experimental evolution of a multiple drug-resistant *P. aeruginosa* clinical isolate toward WCK 5222 resistance.** A clinical *P. aeruginosa* isolate (designated NKPa-71) from a patient with pneumonia is resistant to a variety of antibiotics, including aminoglycosides (tobramycin, gentamicin), fluoroquinolone (ciprofloxacin), β-lactams (meropenem, cefepime, and ceftazidime), and a BLBLI combination ceftazidime-avibactam, but remains susceptible to zidebactam and WCK 5222 (Table 1) (15), leaving WCK 5222 as one of the few choices for the treatment. Whole-genome sequencing (WGS) revealed the presence of multiple antibiotic resistance genes and a 407,629-bp megaplasmid (designated pXP40m) that carries the metallo-β-lactamase gene $bla_{DIM-2}$ (Fig. S1; Table S1).

To examine whether pXP40m is self-transmissible, we performed conjugation between NKPa-71 and PAO1. The transconjugants were verified by PCR amplification of the *parB*, *blaDIM-2*, *blaOXA-1*, and *parM* genes that are localized on pXP40m by using the isolated plasmid as the template (Fig. S2A). The conjugation efficiency was determined in five independent experiments, which was in the range of $1 \times 10^{-8}$ to $4 \times 10^{-8}$ (Fig. S2B). The minimum inhibitory concentrations (MICs) of meropenem, cefepime, ceftazidime, and ceftazidime-avibactam for the transconjugant were 16- to 64-fold higher than those for the parental strain PAO1 (Table 1). However, the MICs of zidebactam and WCK 5222 remain the same as PAO1 (Table 1).

**TABLE 2** Antibiotic susceptibilities and PBP2 mutations of the evolved strains[a]

| Strain | Mutation sites in PBP2 | MIC (μg/mL) | | | | | | | | | |
|---|---|---|---|---|---|---|---|---|---|---|---|
| | | WCK 5222 | ZID | FEP | CAZ | CZA | MEM | TB | GM | CIP | PB |
| NKPa-71 | | 8 | 8 | 256 | 128 | 128 | 16 | 64 | 128 | 16 | 0.25 |
| P1D3 | I450V | 64 | 64 | 256 | 128 | 128 | 32 | 128 | 256 | 16 | 0.5 |
| P1D5 | D351A | 128 | 256 | 512 | 128 | 128 | 16 | 128 | 128 | 16 | 0.5 |
| P2D3 | V598E | 64 | 64 | 256 | 256 | 256 | 16 | 128 | 128 | 8 | 0.5 |
| P2D8 | V598E, G528V | 256 | 256 | 512 | 128 | 128 | 32 | 128 | 256 | 16 | 0.5 |
| P3D7 | V516L | 256 | 256 | 256 | 128 | 128 | 16 | 128 | 256 | 8 | 0.25 |
| P4D3 | V516M | 128 | 128 | 256 | 128 | 128 | 32 | 64 | 128 | 16 | 0.5 |
| P5D4 | V517G | 32 | 32 | 256 | 128 | 128 | 64 | 128 | 128 | 8 | 0.25 |
| P5D9 | V517G, V527G | 128 | 128 | 512 | 256 | 128 | 128 | 128 | 256 | 16 | 0.5 |

[a]MIC, minimum inhibitory concentration; ZID, zidebactam; FEP, cefepime; CAZ, ceftazidime; CZA, ceftazidime-avibactam; MEM, meropenem; TB, tobramycin; GM, gentamicin; CIP, ciprofloxacin; PB, polymyxin B.

We next investigated the role of the pXP40m carried DIM-2 in bacterial resistance to $\beta$-lactam antibiotics. Expression of the $bla_{DIM-2}$ gene in the reference strain PAO1 increased the MICs of meropenum, cefepime, ceftazidime, and ceftazidime-avibactam by more than 16-fold (Table 1). However, the MIC of zidebactam or WCK 5222 was not affected by the expression of the $bla_{DIM-2}$ gene (Table 1). These results indicate that DIM-2 plays an important role in the bacterial resistance to the $\beta$-lactams and ceftazidime-avibactam, and the antimicrobial effect of WCK 5222 to the DIM-2 producing strains is mainly attributed to the zidebactam-mediated inhibition on PBP2.

To examine the potential evolutionary trajectories of NKPa-71 that can lead to WCK 5222 resistance, five biological replicates of NKPa-71 were allowed to evolve in parallel in the absence or presence of WCK 5222. The MICs of the passaged populations were determined each day, and the passaging was repeated until the MIC of WCK 5222 for each of the replicates reached at least 128 μg/mL (Fig. S3A). We then isolated single colonies from the replicates at the MIC upshifting points (e.g., changed from 64 to 128 μg/mL) for MIC verification (Table 2), fitness cost, collateral sensitivity and virulence evaluation. The isolates were designated based on the parallel group numbers and the time of isolation, e.g., P1D5 indicates that the strain was isolated from parallel 1 on day 5 (Table 2; Table S3). Compared to the parental strain, P2D3 and P2D8 displayed the most severe growth defect in the LB medium, while P1D3, P1D5, P3D7, P5D4, and P5D9 displayed moderate growth defects (Fig. 1A). In a competition assay with the parental strain, P1D5, P2D8, and P3D7 showed CI values of around 0.01. However, the CI value of P4D3 was approximately 4.0, indicating a growth advantage over the parental strain (Fig. 1B).

All of the evolved strains except P4D3 displayed increased resistance to aminoglycoside antibiotics, and P5D9 was more resistant to meropenem (Table 2). In a mouse acute pneumonia model, the bacterial loads of P1D3, P2D3, P2D8, and P3D7 were lower than the parental strain, whereas P4D3 exhibited a higher bacterial load, indicating increased virulence (Fig. 1C).

**The *pbpA* gene was recurrently mutated in the evolved WCK 5222-resistant mutants.** To understand the resistance mechanisms of the evolved mutants, we sequenced the genomes of the strains that reached the highest MIC in the five parallel repeats, including P1D5, P2D8, P3D7, P4D3, and P5D9. In addition, to understand the mechanisms of the lower levels of resistance, we randomly chose parallels 1 and 5 and sequenced the genomes of the mutants at the MIC upshifting points, including P1D3, P5D2, and P5D4. All of the mutants except P5D2 contain mutations in the *pbpA* gene (Table 2; Table S2). We then sequenced the *pbpA* genes in the resistant mutants of parallels 2 to 4 at each of the MIC upshifting points and found mutations in the *pbpA* gene in all of the mutants except P3D3 (Table 2; Table S3). The *pbpA* mutants were 4- to 32-fold more resistant to zidebactam and WCK 5222. However, MICs of cefepime for the mutants were the same as the parental strain or increased by 2-fold (Table 2).

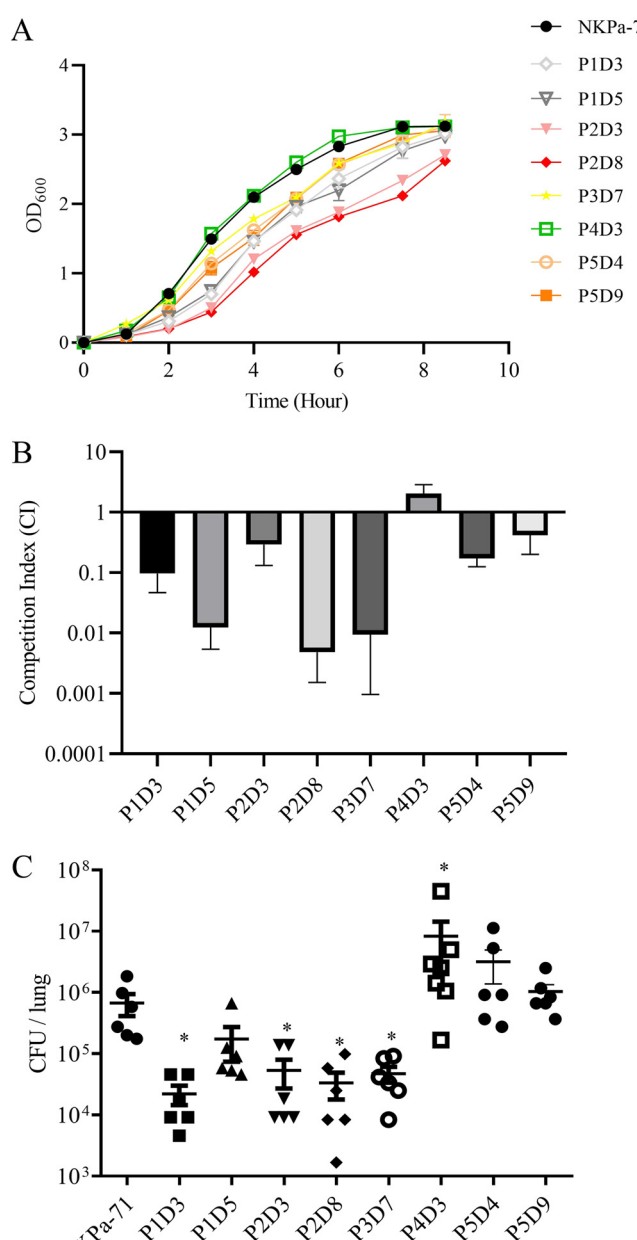

**FIG 1** Growth, competition cost and virulence of the evolved resistant mutants. (A) Bacterial growth rates in LB medium. The same numbers of cells of the NKPa-71 and the evolved mutants were inoculated in LB medium. The bacterial growth was monitored by measuring the optical density at 600 nm ($OD_{600}$) every 1 h. (B) Competition experiment of evolved strains. The competition index is the ratio of the evolved strain to the original strain at 24 h divided by the ratio at 0 h. (C) Bacterial colonization in the murine acute pneumonia model. Each mouse was infected intranasally with $4 \times 10^6$ CFU of NKPa-71 and the evolved resistant mutants. 12 h postinfection, the mice were sacrificed, and the bacterial loads in the lungs were determined. The bars represent medians, and the error bars represent standard deviations. *, $P < 0.05$ by Student's $t$ test.

These results indicate that the evolved resistance to WCK 5222 might be mainly due to the mutations in *pbpA*, which increase the resistance of PBP2 to zidebactam.

**Roles of the PBP2 mutations in zidebactam resistance.** A recent study revealed the structure of *P. aeruginosa* PBP2 in complex with WCK 5153, a DBO that highly resembles the structure of zidebactam (16). By utilizing the structural data, we localized the mutated amino acid residues of PBP2 in a three-dimensional diagram by using PyMOL 2.4.2 (Fig. 2). The mutated residues, namely, D351, I450, V516, V517, V527, G528, and

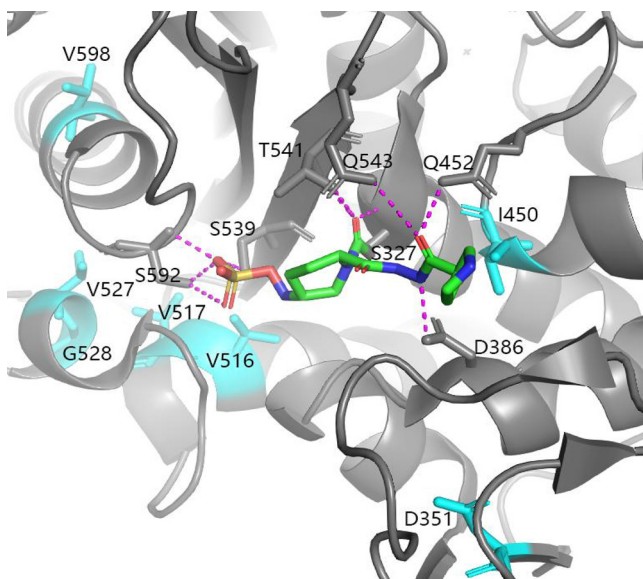

**FIG 2** Structure positioning of the mutated amino acid residues of PBP2. The crystal structure information of *P. aeruginosa* PBP2 in complex with WCK 5153 (Protein Data Bank code 7KIS) was imported into the software PyMOL molecular graphics system (version 2.4.2, Schrödinger, LLC) to localize the positions of the mutated amino acid residues [16]. The mutated amino acid residues of PBP2 in the evolved resistant mutants are colored cyan over a gray background. The bound WCK 5153 is shown in stick representation. Hydrogen bonds are depicted as dashed lines.

V598 are localized in the catalytic domain of PBP2, and I450 was demonstrated to form a van der Waals force with the diacylhydrazide moiety of zidebactam (16).

To understand the roles of the mutations in the resistance to zidebactam, we reconstituted the *pbpA* mutations in wild-type PAO1 and examined their influences on resistance levels (MICs), affinities between PBP2 and zidebactam, and functions of PBP2 (Fig. S4). PBP2 is a transpeptidase that is involved in peptidoglycan cross-linking and cell wall elongation. Inhibition of PBP2 by zidebactam leads to round cell formation (7, 17, 18). In wild-type PAO1, the mean length/width ratio was 3 in LB medium and was reduced to 1 following zidebactam treatment (Table 3). We thus used the length/width ratios to represent the activities of the PBP2 mutants. Based on the phenotypes, the PBP2 mutations could be divided into the following six categories (Fig. S5; Table 3).

**Category 1: the I450V mutation.** The mutation resulted in a 32-fold increase in the MIC. In agreement with the structural result, the I450V mutation increased the dissociation constant between PBP2 and zidebactam by 400-fold (Fig. S4; Table 3). Meanwhile, the mutation led to a constitutive length/width ratio of 1 in the absence and presence of

**TABLE 3** Influences of the PBP2 mutations on resistance levels (MICs), bacterial morphology, and affinity to zidebactam[a]

| Category | Strain | MIC (μg/mL) | | | | | | | | Dissociation constant (×10⁻⁶) | Mean length/width | |
|---|---|---|---|---|---|---|---|---|---|---|---|---|
| | | WCK5222 | ZID | FEP | MEM | CAZ | TB | PB | CIP | | − ZID | + ZID |
| | PAO1 | 4 | 4 | 1 | 0.5 | 2 | 1 | 0.5 | 0.25 | 1.58 | 3 | 1 |
| 1 | I450V | 2 | 128 | 0.5 | 0.5 | 2 | 1 | 1 | 1 | 632.76 | 1 | 1 |
| 2 | D351A | 8 | 256 | 1 | 1 | 4 | 1 | 0.25 | 1 | 14.93 | 1 | 1 |
| | V598E G528V | 8 | 256 | 2 | 0.5 | 4 | 1 | 0.5 | 1 | 9.29 | 1 | 1 |
| 3 | V598E | 2 | 64 | 0.5 | 0.5 | 2 | 1 | 1 | 1 | 22.89 | 2.5 | 1.15 |
| | V527G | 2 | 32 | 1 | 0.5 | 4 | 2 | 1 | 0.5 | 77.19 | 2.5 | 1 |
| 4 | V516M | 8 | 32 | 2 | 0.5 | 1 | 1 | 0.5 | 0.5 | 84.56 | 1.8 | 1.5 |
| | V517G | 4 | 16 | 1 | 0.5 | 1 | 1 | 0.25 | 0.25 | 33.52 | 1.25 | 1.25 |
| | V517G V527G | 4 | 32 | 1 | 0.5 | 1 | 1 | 0.25 | 1 | 65.39 | 1.25 | 1.25 |
| 5 | V516L | 2 | 16 | 1 | 0.5 | 4 | 1 | 0.25 | 2 | 2.33 | 1.5 | 1.75 |
| 6 | G528V | 2 | 4 | 1 | 0.25 | 1 | 1 | 0.25 | 0.5 | 15.09 | 3.3 | 1.5 |

[a]MIC, minimum inhibitory concentration; ZID, zidebactam; FEP, cefepime; MEM, meropenem; CAZ, ceftazidime; TB, tobramycin; PB, polymyxin B; CIP, ciprofloxacin.

zidebactam (Fig. S5; Table 3). These results indicate that the I450V mutation might mainly increase the resistance by reducing the affinity between PBP2 and zidebactam while impairing the function of PBP2.

**Category 2: the D351A and V598E-G528V mutations.** Both of the mutations resulted in the highest increase in the MIC (64-fold) and a constitutive length/width ratio of 1 in the absence and presence of zidebactam (Fig. S5; Table 3). The D351A and V598E-G528V mutations increased the dissociation constants by 9.4- and 5.9-fold, respectively (Fig. S4; Table 3). Thus, the increased resistance might be due to the combined consequences of the moderate reduction in the affinity and functional resistance to zidebactam.

**Category 3: the V598E and V527G mutations.** The mutations resulted in 16- and 8-fold increases in the MIC and 14.5- and 48.9-fold increases in the dissociation constants, respectively (Fig. S4; Table 3). However, the length/width ratios were reduced from 2.5 to 1.15 or 1 by zidebactam (Fig. S5; Table 3). These results indicate that the functions of the V598E and V527G PBP2 mutants might remain susceptible to zidebactam, and the resistance was increased mainly due to the reduced affinity to zidebactam.

**Category 4: the V516M, V517G, and V517G-V527G mutations.** The mutations increased the MIC by 4- to 8-fold and the dissociation constants by 21.2- to 53.5-fold (Fig. S4; Table 3). Compared to the wild-type PAO1, the length/width ratios of the mutants were reduced from 3 to 1.8 or 1.25 in the absence of zidebactam, which, however, were not affected by the treatment of zidebactam (Fig. S5; Table 3). These results indicate that these mutations partially impair the function of PBP2, while rendering the enzyme functionally resistant to zidebactam, which might be due to the reduced affinities.

**Category 5: the V516L mutation.** The mutation increased the MIC by 4-fold. The length/width ratios were 1.5 and 1.75 in the absence and presence of zidebactam, respectively, indicating a functional resistance to the drug (Fig. S5; Table 3). However, the dissociation constant was 1.5-fold of the wild-type PBP2 (Fig. S4; Table 3). Therefore, the V516L PBP2 might remain functional while binding to zidebactam.

**Category 6: the G528V mutation.** The mutation increased the dissociation constant by 9.6-fold (Fig. S4; Table 3). The length/width ratios were 3.3 and 1.5 in the absence and presence of zidebactam, respectively (Fig. S5; Table 3), indicating that the mutation did not affect the normal function of PBP2 and led to a slight functional resistance to zidebactam. The MIC of the G528V mutant remained the same as the wild-type PAO1, which might explain why the G528V mutation was identified with the V598E mutation but did not appear as a single mutation in the passaging assay. However, the G528V mutation further impaired the function of PBP2 and enhanced the zidebactam resistance of the V598E mutant.

**Influences of the PBP2 mutations on cell wall integrity.** Our results demonstrated that the PBP2 mutations resulted in cell rounding to various extents, which might impair cell wall integrity. We thus determined the bacterial survival rates after hypoosmotic shock. The PBP2 V598E-G528V and V527G mutants displayed similar survival rates as the wild-type PAO1. However, the other mutations reduced the bacterial survival rates, among which the D351A mutation resulted in the lowest survival rate (Fig. S6A). In NKPa-71, all of the evolved mutants displayed lower survival rates than the parental strain (Fig. S6B). Compared to the corresponding mutations in PAO1, the D351A and V598E-G528V mutations in NKPa-71 resulted in more severe defects in the resistance to the hypoosmotic shock. The difference might be due to the different genomic backgrounds or the additional mutations in the evolved NKPa-71 mutants.

## DISCUSSION

In this study, we identified a *P. aeruginosa* clinical isolate NKPa-71 that harbors a metallo-$\beta$-lactamase gene $bla_{\text{DIM-2}}$ on a self-transmissible megaplasmid. Parallel evolution experiments demonstrated a quick increase in the MIC of WCK 5222. The clinical breakpoint of WCK 5222 has not been set yet. Two recent studies proposed the tentative breakpoints at $8 + 8$ $\mu$g/mL or $32 + 32$ $\mu$g/mL of cefepime + zidebactam (15, 19). In our tested strains, P5D4 displayed an MIC at $32 + 32$ $\mu$g/mL. The MICs of the other

strains were equal or higher than $64 + 64$ $\mu$g/mL (Table 2), indicating that these strains might cause treatment failure of WCK 5222. All of the WCK 5222-resistant mutants contain missense mutations in the *pbpA* gene that encodes PBP2. In this study, the missense mutations that confer zidebactam resistance range from D351 to V598, which are localized in the transpeptidase domain, and the I450 has been shown to be involved in direct interaction with zidebactam (16).

Among the WCK 5222-resistant mutants of NKPa-71, P1D5, P2D8, and P3D7 that carry D351A, V598E-G528V and V516L mutations displayed CI values around 0.01 against the parental strain and were attenuated in the murine pneumonia model (Fig. 1B and C), which might be due to reduced growth rates (Fig. 1A). Reconstitution of these mutations in PAO1 demonstrated that these mutations reduced the bacterial length/width ratios from 3.0 to 1 or 1.5 (Table 3), indicating impairment of the PBP2 function. Thus, it is likely that strains carrying these mutations might be lost during WCK 5222 treatment in a clinic. In contrast, the V516M mutation resulted in a higher bacterial load than the parental strain. In addition, the V516M mutation was identified in a PAO1 zidebactam-resistant mutant developed in the presence of zidebactam (14). These results indicate that cells carrying the V516M mutation might become prevalent during treatment with WCK 5222 in a clinic.

Previous studies demonstrated that *P. aeruginosa* develop resistance to the BLBLI combination ceftazidime-avibactam due to mutations in the chromosomal *ampC* gene (20, 21). The mutations increase the resistance of AmpC to avibactam and increase its ceftazidime hydrolytic activity (21). In our evolution assay, we did not observe mutation in the *bla*$_{DIM-2}$ gene, presumably because zidebactam does not target DIM-2.

Besides PBP2, we found a R226C mutation in the VgrG6 protein (PA5266) in P1D5 and all of the sequenced strains in parallel 5 (Table S2). The P5D2 strain contains mutations only in the *vgrG6* gene and an intergenic region (Table S2). VgrG6 is a structural protein that is localized at the tip of the H2-T6SS machinery (22). However, reconstitution of the mutation in wild-type PAO1 did not affect the MIC of WCK 5222 (data not shown). It warrants further study to examine whether the mutation increases fitness of the *pbpA* mutants. Overall, our results demonstrated that mutations in the *pbpA* genes play a major role in the development of WCK 5222 in a carbapenem-resistant clinical isolate.

## MATERIALS AND METHODS

**Bacterial strains and growth conditions.** The *Escherichia coli* and *P. aeruginosa* strains used in this study are listed in Table S4. Bacteria were grown at 37°C in the LB medium unless otherwise specified.

To construct the *pbpA* point mutations in wild-type PAO1, the mutated *pbpA* genes with their native promoters were amplified by PCR using the chromosomes of the evolved strains as the template and the primers listed in Table S4. The fragments were ligated into pUC18T-mini-Tn7T-Gm. Integration of each of the *pbpA* fragments into the PAO1 chromosome and deletion of the native *pbpA* gene were performed as previously described (23, 24).

**Whole-genome sequencing and analyses.** Whole-genome sequencing was performed by GENEWIZ Biotechnology Co., Ltd. Bacterial genomic DNA was extracted using a TIANamp bacteria DNA kit (TIANGEN, Beijing, China) and then randomly fragmented to less than 500 bp by sonication (Covaris S220). The fragments were treated with the End Prep Enzyme Mix for end repairing, 5¨ phosphorylation, and dA tailing in one reaction, followed by a T-A ligation to add adaptors to both ends. Then libraries with different indices were multiplexed and loaded onto an Illumina HiSeq/Novaseq instrument (Illumina, San Diego, CA) or a MGI2000 instrument according to the manufacturer's instructions (MGI, Shenzhen, China). The full-length genomes were assembled from long reads obtained from the PacBio RS II system using the HGAP3 pipeline. Antibiotic resistance genes were analyzed using the ResFinder 2.1 server. Core-genome alignment with the *P. aeruginosa* genomes was performed using Parsnp version 1.1.2. MLST was performed following the method on the website http://pubmlst.org/paeruginosa/.

**Experimental evolution to select for WCK 5222-resistant strains.** The continuous WCK 5222 passaging experiment was performed with five repeats in LB medium containing increasing concentrations of WCK 5222 (cefepime-zidebactam at a ratio of 1:1). In day 1, 20-$\mu$L overnight cultures of NKPa-71 were inoculated into four tubes containing 4, 8, 16, and 32 $\mu$g/mL WCK 5222 in 2 mL LB medium, which correspond to 0.5×, 1×, 2× and 4× MIC for NKPa-71, respectively. After 24 h of aerobic incubation at 37°C, cells from the highest concentration of WCK 5222 that allowed bacterial growth to an optical density at 600 nm (OD$_{600}$) of ≥2 were isolated. 20 $\mu$L of the culture was inoculated into 2 mL fresh medium containing WCK 5222 at four increased concentrations, e.g., 1×, 2×, 4×, and 8× MIC. The serial passaging

was repeated for 8 to 10 iterations until the MIC of the evolved strains rose to 128 $\mu$g/mL or higher. Meanwhile, another five repeats passaged in LB medium were used as controls.

**Microscale thermophoresis (MST) assay.** Purified C-terminal His$_6$-tagged PBP2 proteins were dialyzed into a buffer composed of 25 mM HEPES (pH 7.5), 150 mM NaCl, and 0.05% (vol/vol) Tween 20. Then the proteins were labeled with the Monolith NT protein labeling kit (NanoTemper Technologies) according to the manufacturer's instruction. The labeled PBP2 protein (10 nM) was incubated with 0.3 to 10 $\mu$M zidebactam for 20 min at room temperature. Then, the samples were loaded into capillaries and analyzed via a NanoTemper Monolith NT.115 Pico (NanoTemper Technologies GmbH) at 25°C. The LED power and the laser power were set to 60 and 10%, respectively. The software MO Affinity Analysis, version 2.2.2 (NanoTemper Technologies), was used to normalize the fluorescence signal and fit the Hill equation. The whole procedure was performed three times for each sample.

**Ethics statements.** All of the animal studies in this study were performed in compliance with the National and Nankai University guidelines for the use of animals in research. The experiment protocol was approved by the animal care and use committee of the College of Life Sciences of Nankai University (permission number NK-04-2012).

**Mouse acute pneumonia infection model.** The mouse acute pneumonia infection was performed as previously described (25). Bacteria were grown to an OD$_{600}$ of 1.0, then harvested by centrifugation, and adjusted to $2 \times 10^8$ CFU/mL in phosphate-buffered saline (PBS). Each 6- to 8-week-old female BALB/c mouse was anesthetized with an intraperitoneal injection of 100 $\mu$L 7.5% chloral hydrate. A total of 20 $\mu$L of the bacterial suspension was intranasally inoculated into each mouse, resulting in $4 \times 10^6$ CFU/mouse. Six mice were infected with each strain. At 12 h postinfection (hpi), the mice were sacrificed by CO$_2$. The lungs were isolated and homogenized in 1% proteose peptone (Solarbio, Beijing, China). The bacterial loads were determined by plating.

**Hypoosmotic shock survival assay.** The hypoosmotic shock survival assay was performed as previously described (26). Bacteria were grown in LB medium to an OD$_{600}$ of 1.0. A 1-mL sample of the bacteria was collected by centrifugation at $12,000 \times g$ for 1 min and then washed once with 150 mM NaCl. The bacterial pellet was resuspended in 500 $\mu$L 150 mM NaCl or distilled water, followed by incubation at room temperature for 15 min. The live bacteria number was determined by serial dilution with LB medium and plating on LB agar plates.

**Conjugation assay.** The conjugation frequency was measured following previously reported methods with modifications (27, 28). A P$_{prtR}$-lacZ transcriptional fusion and a gentamicin-resistant gene *aacC1* were integrated into the chromosome of PAO1 by using the miniTn7T-Gm vector as previously described (23, 29). The resultant strain was used as the recipient strain. The donor strain NKPa-71 and the recipient strain were grown to an OD$_{600}$ of 1.0 at 37°C and then harvested by centrifugation at $12,000 \times g$ for 2 min. The two strains were mixed at a ratio of 1:1 on a filter membrane, which was placed on the Nutrient-Agar (BD, Difco) plate without antibiotic at 37°C for 2 h. The bacterial mixture was washed into LB medium and plated onto a LB agar plate containing 40 $\mu$g/mL X-Gal, 200 $\mu$g/mL gentamicin, and 8 $\mu$g/mL ceftazidime. After 24 h, the number of blue colonies on the plates was counted and divided by the number of the donor strain to obtain the conjugation frequency. The blue colonies were picked and purified by streaking. The presence of pX40m in the transconjugants was verified by plasmid isolation with a large plasmid extraction kit (OMEGA BioTek, Beijing, China), followed by detection of genes carried by pXP40m by PCR with the primers listed in Table S4.

**Antimicrobial susceptibility test.** The MIC of selected antibiotics were determined by the standard serial 2-fold dilution method in cation-adjusted Mueller-Hinton broth (CAMHB) following the Clinical and Laboratory Standards Institute (CLSI) guidelines (30). All of the MIC tests were performed in triplicate. Zidebactam was purchased from MCE (MedChemExpress, China), and the other antibiotics were purchased from Macklin (China).

**Bacterial morphology observation.** Bacteria were grown to an OD$_{600}$ of 1.0. The bacteria were kept growing in LB medium or treated with 40 $\mu$g/mL zidebactam at 37°C for 15 min. Then the bacterial morphology was observed with light microscopy. The length and width of each individual bacterium were measured with CellSens Dimension (Olympus, Japan). The presented length-to-width ratio of each strain was calculated by averaging the ratios from 45 cells in 9 fields.

**Competition assay.** To distinguish the original strain and evolved isolates, a P$_{prtR}$-lacZ transcriptional fusion was integrated into the chromosome of NKPa-71 by using the miniTn7T-Gm vector as previously described, resulting in NKPa-71-lacZ (23, 29). Competition experiments were performed as previously described with minor modifications (31, 32). $10^6$ CFU of NKPa-71-lacZ was mixed 1:1 with each individual evolved strains in 10 mL LB medium and grown at 37°C for 24 h with agitation at 200 rpm. The cultures were then diluted and plated onto LB agar plates containing 40 $\mu$g/mL X-Gal. After incubation at 37°C overnight, the white and blue colonies were counted as the evolved strains and NKPa-71-lacZ, respectively. Four independent competition experiments were performed for each pair. The competition index (CI) was calculated as the ratio between the evolved strain and NKPa-71-lacZ at *t1* (24 h) divided by the corresponding ratio at time zero.

**Data availability.** The genome sequences were deposited in the NCBI database (SRR16282264 to SRR16282273).

## SUPPLEMENTAL MATERIAL

Supplemental material is available online only.
**SUPPLEMENTAL FILE 1**, PDF file, 1.3 MB.
**SUPPLEMENTAL FILE 2**, XLSX file, 0.01 MB.

## ACKNOWLEDGMENTS

This work was supported by National Key Research and Development Project of China grants 2021YFE0101700 and 82061148018; National Science Foundation of China grants 32170177, 32170199, 31970680, 31970179, and 31870130; Fundamental Research Funds for the Central Universities grant 63213121; and Tianjin Municipal Science and Technology Commission grant 19JCYBJC24700. The funders had no role in study design, data collection and interpretation, or the decision to submit the work for publication.

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
