## [Reviewer comments · Microbiology Spectrum]

Microbiology Spectrum

Molecular Characterization of WCK 5222 (cefepime/zidebactam) Resistant Mutants Developed from a Carbapenem Resistant *Pseudomonas aeruginosa* Clinical Isolate

Xiaolei Pan, Xinrui Zhao, Yuqin Song, Huan Ren, Zhenyang Tian, Qi'an Liang, Yongxin Jin, Fang Bai, Zhihui Cheng, Jie Feng, and Weihui Wu

Corresponding Author(s): Weihui Wu, Nankai University

Review Timeline:

Submission Date:	January 4, 2022
Editorial Decision:	January 23, 2022
Revision Received:	January 26, 2022
Accepted:	January 27, 2022

Editor: Beile Gao

Reviewer(s): The reviewers have opted to remain anonymous.

Transaction Report:

DOI: <https://doi.org/10.1128/spectrum.02678-21>

January 23, 2022

Dr. Weihui Wu
Nankai University
State Key Laboratory of Medicinal Chemical Biology, Key Laboratory of Molecular Microbiology and Technology of the Ministry of Education, Department of Microbiology, College of Life Sciences
94 Weijin Road
Nankai District
Tianjin 300071
China

Re: Spectrum02678-21 (Molecular Characterization of WCK 5222 (cefepime/zidebactam) Resistant Mutants Developed from a Carbapenem Resistant *Pseudomonas aeruginosa* Clinical Isolate)

Dear Dr. Weihui Wu:

Link Not Available

Sincerely,

Beile Gao

Journals Department
Reviewer comments:

Reviewer #1 (Comments for the Author):

In this manuscript, Pan et.al., isolated a multidrug resistant *P. aeruginosa* clinical strain called NKPa-71, which is resistant to a variety of β -lactam antibiotics and ceftazidime-avibactam combination. A self-transmissible megaplasmid was identified in this strain, which harbored a metallo- β -lactamase gene blaDIM-2, conferred this strain resistance to β -lactam antibiotics, except WCK 5222. By in vitro passaging assay and whole genome sequencing the authors characterized substantial mutations in zidebactam target protein PBP2 encoding gene *pbpA*, and the authors verified the the roles of these mutations in reference strain PAO1, revealed that mutations in the *pbpA* gene might be a major cause of evolved resistance to WCK 5222 in clinical settings. The study is well designed and conducted, with a large number of solid experiments. The authors uncovered significant information about the underlying resistance mechanisms.

Minor Comments

1. I found that the authors did not present more discussion about the results that mutations of PBP2 caused different effect on bacterial growth as well as competition index (CI) in competition assay, what they thought might be the physiological significance.

2. A number of grammatical errors need corrected (a few examples below):

Line 20: bacteria-> bacterial

Line 22 and 86: resistance-> resistant

Line 78:mutation-> mutations

Line 98 : and-> to

Line 195 : of that of-> of

Reviewer #2 (Comments for the Author):

The authors have addressed previous comments.

Staff Comments:

Preparing Revision Guidelines

Please return the manuscript within 60 days; if you cannot complete the modification within this time period, please contact me. If you do not wish to modify the manuscript and prefer to submit it to another journal, please notify me of your decision immediately so that the manuscript may be formally withdrawn from consideration by Microbiology Spectrum.

Review comments

In this manuscript, Pan et.al., isolated a multidrug resistant *P. aeruginosa* clinical strain called NKPa-71, which is resistant to a variety of β -lactam antibiotics and ceftazidime-avibactam combination. A self-transmissible megaplasmid was identified in this strain, which harbored a metallo- β -lactamase gene blaDIM-2, conferred this strain resistance to β -lactam antibiotics, except WCK 5222. By in vitro passaging assay and whole genome sequencing the authors characterized substantial mutations in zidebactam target protein PBP2 encoding gene *pbpA*, and the authors verified the the roles of these mutations in reference strain PAO1, revealed that mutations in the *pbpA* gene might be a major cause of evolved resistance to WCK 5222 in clinical settings. The study is well designed and conducted, with a large number of solid experiments. The authors uncovered significant information about the underlying resistance mechanisms.

Minor Comments

1. I found that the authors did not present more discussion about the results that mutations of PBP2 caused different effect on bacterial growth as well as competition index (CI) in competition assay, what they thought might be the physiological significance.

2. A number of grammatical errors need corrected (a few examples below):

Line 20: bacteria-> bacterial

Line 22 and 86: resistance-> resistant

Line 78:mutation-> mutations

Line 98: and-> to

Line 195: of that of-> of

Dear Editor:

Please accept my gratitude, to both you and your reviewers, for the attention given to our manuscript entitled “Molecular Characterization of WCK 5222 (cefepime/zidebactam) Resistant Mutants Developed from a Carbapenem Resistant *Pseudomonas aeruginosa* Clinical Isolate” [Spectrum02678-21]. The following is a point-by-point synopsis of the changes we have made to address the comments and concerns.

Reviewer #1 (Comments for the Author):

In this manuscript, Pan et.al., isolated a multidrug resistant *P. aeruginosa* clinical strain called NKPa-71, which is resistant to a variety of β -lactam antibiotics and ceftazidime-avibactam combination. A self-transmissible megaplasmid was identified in this strain, which harbored a metallo- β -lactamase gene blaDIM-2, conferred this strain resistance to β -lactam antibiotics, except WCK 5222. By in vitro passaging assay and whole genome sequencing the authors characterized substantial mutations in zidebactam target protein PBP2 encoding gene pbpA, and the authors verified the the roles of these mutations in reference strain PAO1, revealed that mutations in the pbpA gene might be a major cause of evolved resistance to WCK 5222 in clinical settings. The study is well designed and conducted, with a large number of solid experiments. The authors uncovered significant information about the underlying resistance mechanisms.

Minor Comments

1. I found that the authors did not present more discussion about the results that mutations of PBP2 caused different effect on bacterial growth as well as competition index (CI) in competition assay, what they thought might be the physiological significance.

Response: We added the discussion about the effects of the PBP2 mutations on bacterial growth and competition index (lines 232-239).

2. A number of grammatical errors need corrected (a few examples below):

Line 20: bacteria-> bacterial

Line 22 and 86: resistance-> resistant

Line 78: mutation-> mutations

Line 98: and-> to

Line 195: of that of-> of

Response: The errors have been corrected. We also double checked the whole manuscript to correct grammatical errors.

Reviewer #2 (Comments for the Author):

The authors have addressed previous comments.

My coauthors and I agree that the changes made to the manuscript in addressing the reviewers' concerns have greatly improved this manuscript. Please contact me if I can provide any additional information.

Sincerely,

Weihui Wu, Ph.D.

January 27, 2022

Dr. Weihui Wu
Nankai University
State Key Laboratory of Medicinal Chemical Biology, Key Laboratory of Molecular Microbiology and Technology of the Ministry of Education, Department of Microbiology, College of Life Sciences
94 Weijin Road
Nankai District
Tianjin 300071
China

Re: Spectrum02678-21R1 (Molecular Characterization of WCK 5222 (cefepime/zidebactam) Resistant Mutants Developed from a Carbapenem Resistant *Pseudomonas aeruginosa* Clinical Isolate)

Dear Dr. Weihui Wu:

Your manuscript has been accepted, and I am forwarding it to the ASM Journals Department for publication. You will be notified when your proofs are ready to be viewed.

Sincerely,

Beile Gao
Editor, Microbiology Spectrum

Journals Department
Supplemental table 2: Accept
Supplemental Materials: Accept